

**The role of the winter residual circulation in the summer mesopause**
**regions in WACCM**
Maartje Sanne Kuilman[1], Bodil Karlsson[1]
[1]Department of Meteorology, Stockholm University, 10691 Stockholm,
Sweden.
*Correspondence to:* Maartje Kuilman (maartje.kuilman@misu.su.se)
**Abstract**
High winter planetary wave activity warms the summer polar mesopause via a
link between the two hemispheres. In a recent study carried out with the
Kühlungsborn Mechanistic general Circulation Model (KMCM), it was shown
that the net effect of this interhemispheric coupling mechanism is a cooling of
the summer polar mesospheres and that this temperature response is tied to
the strength of the gravity wave-driven winter mesospheric flow. We here
reconfirm the hypothesis that the summer polar mesosphere would be
substantially warmer without the circulation in the winter mesosphere, using
the widely-used Whole Atmosphere Community Climate Model (WACCM). In
addition, the role of the stratosphere in shaping the conditions of the summer
polar mesosphere is investigated. Using composite analysis, we show that if
winter gravity waves are absent, a weak stratospheric Brewer-Dobson
circulation would lead to a warming of the summer mesosphere region instead
of a cooling, and vice versa. This is opposing the temperature signal of the
interhemispheric coupling in the mesosphere, in which a cold winter
stratosphere goes together with a cold summer mesopause. We hereby
strengthen the evidence that the equatorial mesospheric temperature
response, driven by the winter gravity waves, is a crucial step in the
interhemispheric coupling mechanism.
**1 Introduction**

The circulation in the mesosphere is driven by atmospheric gravity waves.
These waves originate from the lower atmosphere and as they propagate
upwards, they are filtered by the zonal wind in the stratosphere (e.g. Fritts and
Alexander, 2003). Because of the decreasing density with altitude and as a



result of energy conservation, the waves grow in amplitude. At certain
altitudes, the waves – depending on their phase speeds relative to the
background wind - become unstable and break. At the level of breaking, the
waves deposit their momentum into the background flow, creating a drag on
the zonal winds in the mesosphere, which establishes the pole-to-pole
circulation (e.g. Lindzen, 1981; Holton, 1982,1983; Garcia and Solomon,
1985). This circulation drives the temperatures far away from the state of
radiative balance, by adiabatically heating the winter mesopause and
adiabatically cooling the summertime mesopause (Andrews et al., 1987;
Haurwitz, 1961; Garcia and Solomon, 1985; Fritts and Alexander, 2003). The
adiabatic cooling in the summer leads to temperatures sometimes lower than
130 K in the summer mesopause (Lübken et al.,1990). These low
temperatures allow for the formation of thin ice clouds in the summer
mesopause region, the so-called noctilucent clouds (NLCs).

Previous studies have shown that the summer polar mesosphere is influenced
by the winter stratosphere via a chain of wave-mean flow interactions (e.g.
Becker and Schmitz, 2003; Becker et al., 2004; Karlsson et al., 2009). This
phenomenon, termed interhemispheric coupling (IHC), manifests itself as an
anomaly of the zonal mean temperatures. Its pattern consists of a quadrupole
in the winter hemisphere with a warming (cooling) of the polar stratosphere
and an associated cooling (warming) in the equatorial stratosphere. Above in
the mesosphere the temperature anomaly field is reversed with a cooling
(warming) on top of the stratospheric warming (cooling) in the polar
mesosphere, and an associated warming (cooling) in the equatorial region.
The mesospheric warming (cooling) in the tropical region extends to the
summer mesopause (see e.g. Körnich and Becker, 2010).

The IHC pattern was first found using mechanistic models (Becker and
Schmitz, 2003; Becker et al., 2004; Becker and Fritts, 2006), underpinned by
observations of mesospheric conditions. The pattern was then found in
observational data (e.g. Karlsson et al., 2007; Gumbel and Karlsson, 2011;
Espy et al., 2011: de Wit et al., 2016), in the Whole Atmosphere Community
Climate Model (WACCM: Sassi et al. 2004, Tan et al., 2012), in the Canadian





Middle Atmosphere Model (CMAM: Karlsson et al. 2009), and in the high
altitude analysis from the Navy Operational Global Atmospheric Prediction
System- Advanced Level Physics High Altitude (NOGAPS-ALPHA)
forecast/assimilating system (Siskind et al., 2011).

The anomalies in the zonal-mean temperature fields are responses to
different wave forcing in the winter hemisphere. A stronger planetary wave
forcing in the winter stratosphere yields a stronger stratospheric Brewer-
Dobson circulation (BDC). This anomalously strong flow yields an
anomalously cold stratospheric tropical region and a warm stratospheric
winter pole, due to the downward control principle (Haynes et al. 1991). The
mechanism discussed here is for the case of a stronger winter residual
circulation, but works the same for a weakening (Karlsson et al., 2009).

Due to the eastward zonal flow in the winter stratosphere, GWs carrying
westward momentum propagate relatively freely up through the mesosphere
where they break. Therefore, in the winter mesosphere, the net drag from
GWs momentum deposition is westward. When vertically propagating
planetary waves break – also carrying westward momentum – in the
stratosphere, the momentum deposited onto the mean flow decelerates the
stratospheric westerly winter flow. To put it short, a weaker zonal
stratospheric winter flow allows for the upward propagation of more GWs with
an eastward phase speed, which, as they break reduces the westward wave
drag (see Becker and Schmitz, 2003, for a more rigorous description). This
filtering effect of the zonal background flow on the GW propagation results in
a reduction in strength of the winter-side mesospheric residual circulation
when the BDC is stronger. The downward control principle now causes the
mesospheric polar winter region to be anomalously cold and the tropical
mesosphere to be anomalously warm (Becker and Schmitz, 2003, Becker et
al., 2004; Körnich and Becker, 2009).

The critical step for IHC is the crossing of the temperature signal over the
equator. The essential region is here the equatorial mesosphere. Central in
the hypothesis of IHC is that the increase (or decrease) of the temperature in




the tropical mesosphere modifies the temperature gradient between high and
low latitudes in the summer mesosphere, which influences the zonal wind in
the summer mesosphere, due to thermal wind balance (see e.g. Karlsson et
al., 2009 and Karlsson and Becker, 2016).
The zonal wind change in the summer mesosphere modifies the breaking
level of the summer-side GWs. In the case of a warming in the equatorial
mesosphere – as when the BDC is strong -, the zonal wind is modified in such
a way that the intrinsic wave speeds are reduced (e.g. Becker and Schmitz,
2003; Körnich and Becker, 2009). When the relative speed between the GWs
and the zonal flow decreases, the GWs break at a lower altitude, thereby
shifting down the GW drag per unit mass. The upper branch of the residual
circulation also shifts downwards and along with this shift there is a reduction
of adiabatic cooling, which causes a positive temperature anomaly in the
summer mesosphere (Karlsson et al., 2009; Körnich and Becker, 2009;
Karlsson and Becker, 2016). In the case of an equatorial mesospheric cooling,
the response is the opposite: the relative difference between the zonal flow
and the phase speeds of the gravity waves increase to that they break at a
slightly higher altitude, with a anomalous cooling of the summer mesopause
as a result.

The interhemispheric coupling mechanism is debated. For example,
Pendlebury (2012) and Siskind and McCormack (2014) suggest the quasi-2
day (Q2DW) wave to be involved in transferring the signal from the equatorial
region to the summer polar mesopause region. They show that enhanced
Q2DW activity leads to a warming of the summer mesopause. We argue that
the Q2DW is an additional mechanism that comes into play controlling the
summer mesospheric temperatures, adding to the effects of the IHC
mechanism. A strong indication of it being two separate mechanisms – not
necessarily unconnected – was presented by Karlsson and Becker (2016),
who showed, using the Kühlungsborn Mechanistic general Circulation Model
(KMCM), a more fundamental role of the interhemispheric coupling; the
mechanism has a net cooling effect on the summer polar mesosphere. IHC
has hitherto primarily been seen as a mode of internal variability giving rise to
a warming of the summer polar mesopause region.





As mentioned above, the equatorial mesosphere is of crucial importance for
interhemispheric coupling. The temperature in this region is modified by the
strength of the residual circulation in the winter mesosphere. Karlsson and
Becker (2016) hypothesized that if the GW-driven winter residual circulation
would not be present, the equatorial mesosphere would be warmer, which
would lead to lower breaking levels of GWs and a warmer summer
mesosphere region. Analogically, an anomalously cold equatorial region
would lead to an anomalously cold summer mesosphere region (e.g. Karlsson
et al., 2009; Karlsson and Becker, 2016).

Becker and Karlsson (2016) showed that the equatorial mesosphere is
substantially colder in July than it is in January, while the winter mesosphere
is significantly warmer (see their Fig. 1). That means that the GWs break
higher in the NH summer mesosphere than in the SH summer mesosphere,
which is one possible reason for why the July summer polar mesosphere is
colder than in the January summer polar mesosphere (e.g. Becker and Fritts,
2006; Karlsson et al., 2009). If – as hypothesized by Karlsson and Becker
(2016) – the fundamental effect of the IHC is a cooling of the summer
mesopauses, it would mean that the mechanism plays a more important role
affecting the temperatures in the summer mesopause in the NH compared to
that in the SH, since the weaker planetary wave activity in the SH results in an
increased gravity wave drag and a strengthening of mesospheric poleward
flow in the winter mesosphere. The equatorial mesosphere would then be
adiabatically cooled more efficiently than when the winter mesospheric
circulation is weak. In the same manner, the NH winter has, in a climatological
sense, a weaker effect on the residual circulation in the SH summer
mesosphere, according to the mechanism described before.

Karlsson and Becker (2016) hypothesized that in the absence of the equator-
to-pole flow in the SH winter, the summer mesopause in the NH would be
considerably warmer.  Moreover, removing the mesospheric residual
circulation in the NH winter would not have as high impact on the SH summer
mesopause. To test the hypothesis, they used the KMCM to compare control





simulations to runs without GWs in the winter mesosphere. The predicted
responses were confirmed, and the results were also backed up by correlation
studies using the Canadian Middle Atmosphere Model (CMAM30).

Since IHC is controversial, we find it important to use as many tools as
possible to test – and to underpin - our arguments. In this study, the widely-
used WACCM, described in Section 2.1 below, is used to endorse the results
obtained with the not as widely-used – yet comprehensive – KMCM. To
investigate the consequences for noctilucent clouds, formed in the
mesopause region, of removing the winter mesospheric residual flow, we
implement a basic cloud parameterization, as described in Section 2.2. The
Whole Atmosphere Community Climate Model (WACCM) results from
comparing runs with and without winter GWs are presented in Section 3. As
an important complement to the study carried out by Karlsson and Becker
(2016), we here examine the role of the summer stratosphere in shaping the
conditions of the NH summer polar mesosphere when the winter mesospheric
flow is absent. We focus on the effect that the zonal wind in the summer
stratosphere has, and study if and how the PW activity in the winter affects
the summer polar mesosphere. These results are presented in Section 3.1.
Our conclusions are summarized in Section 4. Since the IHC mechanism has
a more robust signal in the SH winter – NH summer, we choose to focus
particularly on this period, namely July. Nevertheless, results from January
are also shown for comparisons and for further discussion.
**2 Method**

**2.1 Model**

The Whole Atmosphere Community Climate Model (WACCM) is a so-called
"high-top" chemistry-climate model, which spans the range of altitude from the
Earth's surface to an altitude of about 140 km.  WACCM has 66 vertical levels
of a resolution of ~1.1 km in the troposphere above the boundary layer, 1.1-
1.4 km in the lower stratosphere, 1.75 km at the stratosphere and 3.5 km





above 65 km. The horizontal resolution is 1.9° latitude by 2.5° longitude
(Marsh et al, 2013).

The model is a component of the Community Earth System Model (CESM),
which is a group of model components at the National Center for Atmospheric
Research (NCAR). WACCM is a superset of the Community Atmospheric
Model version 4 (CAM4) and as such it includes all the physical
parameterizations of CAM4 (Neale et al., 2013).

WACCM includes parameterized non-orographic gravity waves, which are
generated by frontal systems and convection (Richter et al., 2010). The
orographic GW parameterization is based on McFarlane (1987), while the
nonorographic GW propagation parameterization is based the formulation by
Lindzen (1981).

In this study, The F_2000_WACCM (FW) compset of the model is used, i.e.
the model assumes present day conditions. There is no forcing applied: the
model runs a perpetual year 2000. Our results are based on a control run and
perturbation runs. In the control run, the winter side residual circulation is
included. In the perturbation runs, the equator-to-pole flow is removed by
turning off both the orographic and the non-orographic gravity waves. It
should however be noted that even though the GWs are turned off, there are
still some resolved waves, such as inertial gravity waves and planetary waves
that drive a weak meridional circulation. The model is run for 30 years.

**2.2 Noctilucent clouds**
It was discussed earlier that the gravity-wave driven residual circulation in the
middle atmosphere causes the temperatures in the summer mesopause
region to be extremely low (e.g. Andrews et al., 1987), which allows for the
formation of noctilucent clouds (NLCs) in this region. In the northern
hemisphere, a typical NLC season lasts from late May until the end of August.
In the southern hemisphere, the NLCs are present from the end of November
until mid-February (e.g. Thomas and Olivero, 1989).



We parameterize these clouds in WACCM using the temperature and water
vapor. We calculate the ice mass, assuming that water vapour can turn into
ice if its partial pressure is larger than the saturation pressure. The saturation
pressure is calculated using a fit to the numerical solution of the Clausius-
Clapeyron equation, as derived by Murphy and Koop (2005). This model is
based on the approach of Hervig et al. (2009).
Our method assumes that the ice exists in local thermodynamic equilibrium.
This assumption has been shown to lead to an overestimation of the ice mass
(e.g. Rong et al., 2010). Therefore, we assume that half of the water goes into
ice, following a recent study by Christensen et al. (2016). We do not account
for microphysical processes, as it has been shown before that NLCs can be
modeled with very limited knowledge of their nucleation properties (Merkel et
al., 2009; Megner et al., 2011).
**3 Results and discussion**
To investigate the effect of the winter residual circulation on the summer
mesopause, we compare the control run, which includes the winter equator-
to-pole circulation, with the perturbation runs. In the perturbation runs, the
equator-to-pole flow is removed by turning off the parameterized gravity
waves. The resolved waves, such as tides, inertial gravity waves and
planetary waves are still there and drive a weak poleward flow, as already
described in section 2.1.
We start by investigating the case for the NH summer (July) with the GWs
turned off for the SH, where it is winter. Figure 1 shows the difference in
zonal-mean temperature and zonal-mean zonal wind for July as a function of
latitude and altitude, between the control run and the perturbation run: the run
without the GWs in the winter minus the run with the GWs in the SH.
Figure 1.
From Fig. 1, it is clear that there is a considerable increase in temperature in
the NH summer mesopause region in the case for which there is no equator-
to-pole flow in the SH winter. Without the GWs in the SH winter, the winter



stratosphere and lower mesosphere are colder. This can be understood as
GWs in the winter hemisphere drive downwelling, which adiabatically heats
these regions. It is also clear that the zonal flow at high latitudes accelerates
for the case for which there is no equator-to-pole flow in the SH winter. These
findings correspond with what is found in Karlsson and Becker (2016).
It can also be seen that like in the KMCM model, the zonal wind and
temperature in summer stratosphere region change only slightly in the
perturbation runs as compared to the control runs. We deem that anomalous
GW filtering effects from the lower down in the summer stratosphere, which
could affect the results, are unlikely to contribute substantially to the
temperature change in the summer mesosphere. We come back to this
question in the next paragraph 3.1.
There is less upwelling in the NH summer mesopause in the case where the
GWs in the SH winter hemisphere are turned off. We have seen that this
leads to an increase in temperature in the summer mesopause, but at the
same it leads to a decrease in water vapor concentration in the same region,
as can be seen in Fig. 2. As a result of the increased temperature and
decreased water vapor concentration, the noctilucent cloud ice mass density
reduces, as is clear from Fig. 2.
Figure 2
The mechanism behind the reduction of the water vapor and the temperature
increase is further illustrated in Fig. 3, which shows the zonal wind between
45°N and 70°N, GW drag and temperature between 70°N and 90°N in July for
the control and perturbation run. As a result of the changed meridional
temperature gradient, the westward jet is weaker in the case in which there
are no GWs in the winter hemisphere. The weaker jet, leads to lower GW
levels and weaker GW drag as can be seen in Fig 3. Figure 3 also shows the
temperature over the latitude bands 70° - 90° N, from this it can be seen that
summer polar mesopause is considerably warmer if there are no GWs in the
winter hemisphere.
Figure 3



To investigate the IHC mechanism further, we also show the correlation and
covariance, which also provides information about the amplitude of the
variability, between the temperature in the winter stratosphere in July (1-10
hPa, 60°S-40°S) and the temperatures in the rest of the atmosphere in the
same month. We show the correlation and covariance fields for both the
cases with and without GWs in the SH winter hemisphere.
Figure 4
In the correlation and covariance fields of the control run, the temperature in
the winter stratosphere is positively correlated with the temperature in the
equatorial mesosphere and the summer mesopause region. If the GWs are
removed in the winter hemisphere, the temperature in the summer
mesopause region anti-correlates with the temperature in the winter
stratosphere. Also, the temperature in the equatorial mesosphere does no
longer correlate and co-vary significantly with the temperature in the winter
hemisphere, in agreement with the results of Karlsson and Becker, 2016.
Until now, we investigated the influence of the SH winter residual circulation
on the NH summer mesopause (in July).  Now, we will also investigate the
effect that the NH winter residual circulation has on the SH summer
mesosphere (in January). We discussed earlier that this effect will be smaller
as compared to the effect of the SH winter residual circulation on the NH
summer mesosphere (in July). Figure 5 shows the difference in zonal-mean
temperature and zonal-mean zonal wind for January as a function of latitude
and altitude, between the control run and the perturbation run:  the run without
the GWs in the NH winter hemisphere minus the run with the GWs in the NH
winter hemisphere.
Figure 5.
From Fig. 5, it can be observed that there is not such a clear increase in
temperature in the SH summer mesopause region in the case for which there
is no equator-to-pole flow in the NH winter.  There is a small increase in the
temperature for the upper part of the SH NLC region (January), but this
change is not statistically significant. Without the GWs in the winter





hemisphere, the winter stratosphere and lower mesosphere are colder, as in
the July case. There is a change in zonal wind at high southern latitudes, but
there is no clear statistical significant increase. These findings correspond
with what is hypothesized: the SH summer is less affected by the IHC
mechanism.
In Fig. 6, we show the correlation and covariance between the temperature in
the winter stratosphere in January (1-10 hPa, 60°S-40°S) and the
temperatures in the rest of the atmosphere in the same month for both the
cases with and without GWs in the NH winter hemisphere.
Figure 6
The general pattern in January for the correlation and covariance for both the
control run and the run without GWs in the winter hemisphere is very similar
to the pattern in July. However, the correlation and covariance in the summer
mesosphere with the temperatures in the winter stratosphere are not
statistically significant. This can be understood, as the variability in the SH
summer mesopause region in January is much higher. It is seen that in both
hemispheres, the temperature in the equatorial mesosphere correlates
statistically significant with the temperatures in the winter stratosphere for the
control case, but not for the case without the GWs in the winter hemisphere.
**3.1 The role of the summer stratosphere region**
In this section, we focus on the effect that the summer stratosphere has on
the summer mesosphere in the absence of a mesospheric winter residual flow.
We investigate if and how the planetary wave (PW) activity in the winter
affects the summer polar mesosphere. We choose to focus particularly on the
NH summer in July. However, we also show the effect of the SH summer
stratosphere on the SH summer mesosphere in January for comparison and
further discussion.

We start by looking at the control case in July, for which the GWs in the winter
hemisphere are on. We use the temperature in the winter stratosphere (1-10
hPa, 60°S-40°S; see Karlsson et al., 2007) as a proxy for the strength of the





Brewer-Dobson circulation and composite strong and weak cases. The
anomalous temperature responses are shown in Fig. 7. It can be seen that
when the temperature in the winter stratosphere region is anomalously low
(high), there is a cooling (warming) of the NLC region.
Figure 7
The cold (warm) winter stratosphere is caused by an anomalously weak
(strong) Brewer-Dobson circulation, which leads to a cooling (warming) of the
equatorial mesosphere. This tropical temperature response changes the
meridional temperature gradient in the summer mesosphere, and thereby –
via thermal wind balance - the zonal mesospheric winds. The zonal wind
change modifies the GW drag in such a way that a cooling (warming) of the
NH summer mesopause is generated (see e.g. Karlsson et al. 2009). We note
that a reversed meridional temperature gradient occurs simultaneously in the
summer stratosphere as a response to the BDC. However, as pointed out by
Karlsson et al. (2009), the expected GW filtering effect of this stratospheric
temperature gradient would oppose that of the mesospheric temperature
gradient.
With the mesospheric winter residual circulation being out of play, it is
straight-forward to investigate effect of the temperature gradient in the
summer stratosphere. Again, we show the anomaly fields for weak and strong
stratospheric residual flow in the SH winter stratosphere (1-10 hPa, 60°S-
40°S) in July, but this time without the winter GWs.
Figure 8
From Fig 8., it is clear that taking away the GWs in the SH winter hemisphere
changes the response to anomalously high or low temperatures (i.e. high and
low PW-activity, respectively: see e.g. Karlsson et al., 2007) in the summer
mesopause region. Anomalously low temperatures in the SH winter
stratosphere, indicating a weak Brewer-Dobson circulation, now lead to a
warming in the NH summer mesopause region, instead of a cooling as
observed in the case where there are GWs in the SH winter hemisphere.





We hypothesize that this opposing signal is – in the absence of a
mesospheric residual flow in the winter - caused by a modulation of the
meridional temperature gradient in the summer stratosphere, inferred by the
BDC.
To strengthen our arguments, we plot the vertical profiles of the zonal wind,
GW drag between 45°N-55°N and the temperatures between 70°N-90°N in
July. These profiles are shown for both high and low temperatures in the
winter stratosphere (1-10 hPa, 60°S-40°S).  The differences between the
cases with anomalously low and high temperatures are also plotted.
Figure 9
From Fig. 9, it is clear for a weak Brewer-Dobson circulation, and therefore
anomalously low temperatures in the SH winter stratosphere, the zonal winds
in the stratosphere are less strongly westwards. This leads to a weaker GW
drag and a warmer NH summer mesopause region.
We hereby suggest that without GWs in the SH winter hemisphere, it would
be the variability in the NH summer stratosphere caused by the winter-side
BDC that would have the major influence on the temperatures in the NH
summer mesopause. A weaker (stronger) Brewer-Dobson circulation would
lead to a change in the temperature gradient in the summer stratopause,
which would lead to a cooling (warming) instead of the warming (cooling)
associated with interhemispheric coupling.
We also discuss the effect of the SH summer stratosphere on the SH summer
mesosphere (in January). Also here, we start by looking at the control case, in
which the GWs in the NH winter hemisphere are on.
We use the temperature in the winter stratosphere (1-10 hPa, 60°N-80°N) in
January as a proxy for the strength of the Brewer-Dobson circulation and
composite strong and weak cases. The anomalous temperature responses
are shown in Fig. 10. In Fig. 6, we saw that the correlation of the temperatures
with the winter stratosphere do not always reach a level of statistical
significance of 95%.  However, from Fig. 10 it is clear that the pattern is the





same as for the case in July: when the temperature in the winter stratosphere
region is anomalously low (high), there is a cooling (warming) of the NLC
region.
Figure 10
Like we did for the July case, we show the anomaly fields for weak and strong
stratospheric residual flow in the winter stratosphere (1-10 hPa, 60°N-80°S) in
January, for the case without the winter GWs.
Figure 11
From Fig 11., it is clear that also for the January, taking away the GWs in the
winter hemisphere leads to a different response to anomalously high or low
temperatures in the winter stratosphere as compared to the control case. As
in the July, anomalously low temperatures in the winter stratosphere
(associated with a weak Brewer-Dobson circulation) lead to a warming in the
summer mesopause region, instead of a cooling for the case where there are
GWs in the winter hemisphere.
In Fig. 12, we show the vertical profiles of the zonal wind, GW drag between
45°S-55°S and the temperatures between 70°S-90°S for both high and low
temperatures in the winter stratosphere (1-10 hPa, 40°N-60°N, January). In
addition, the differences between the cases with anomalously low and high
temperatures are shown.
Figure 12
The profiles for the southern hemisphere in January are very similar to the
profiles for the northern hemisphere in July. Also here, for a weak Brewer-
Dobson circulation, the zonal winds in the stratosphere are less strongly
westwards, leading to a weaker GW drag and a warmer summer mesopause
region. To summarize, both in the northern and summer hemisphere, a
weaker (stronger) Brewer-Dobson circulation leads to a change in the
temperature gradient in the summer stratopause, which leads to a warming
(cooling) instead of the cooling (warming) that is associated with





interhemispheric coupling.
**4 Conclusions**
In this study, the interhemispheric coupling mechanism and the role of the
summer stratosphere in shaping the conditions of the summer polar
mesosphere have been investigated. We have used the widely used WACCM
model to reconfirm the hypothesis of Karlsson and Becker (2016) that the
summer polar mesosphere would be substantially warmer without the gravity
wave-driven residual circulation in the winter. We find, in accordance with the
previous study, that the interhemispheric coupling mechanism has a net
cooling effect on the summer polar mesospheres. We also find that the
mechanism plays a more important role affecting the temperatures in the
summer mesopause in the NH compared to that in the SH.

We have also investigated the role of the summer stratosphere in shaping the
conditions of the summer polar mesosphere. It is shown that without the
winter mesospheric residual circulation, the variability in the summer polar
mesosphere is determined by the temperature gradient in the summer
stratosphere below, which is modulated by the strength of the BDC. We have
found that for both the northern and the southern hemisphere, in the absence
of winter gravity waves, a weak Brewer-Dobson circulation would lead to a
warming of the summer mesosphere region. The temperature signal of the
interhemispheric coupling mechanism is opposite: in this case a weak Brewer-
Dobson circulation, the summer mesosphere region is cooled. This confirms
the idea that it is the equatorial mesosphere that is governing the
temperatures in the summer mesopause regions, rather than processes in the
summer stratosphere.



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





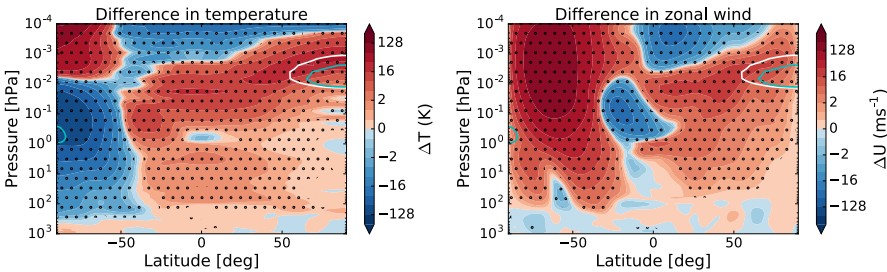

Fig.1. The difference in zonal-mean temperature (left) and zonal-mean zonal wind (right) for July: [run without winter GWs] minus [control run]. The white contour indicates the summer polar mesopause region where the temperatures are below 150 K for the control run. The blue contour indicates the region where the temperature is below 150 K for the run without the GWs in winter. The dotted areas are regions where the data reaches a confidence level of 95%.

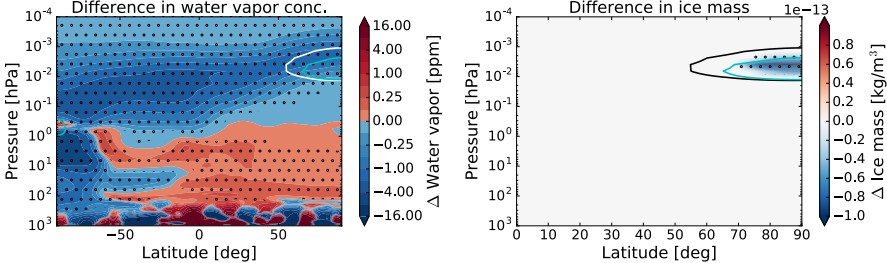

Fig. 2. The difference in zonal-mean water vapour concentration (left) and zonal-mean ice mass density (right) for July: [run without winter GWs] minus [control run]. The black contour indicates the region where the temperatures is below 150 K for the control run. The blue contour indicates the region where the temperature is below 150 K for the run without the GWs in winter. The dotted areas are regions where the data reaches a confidence level of 95%.

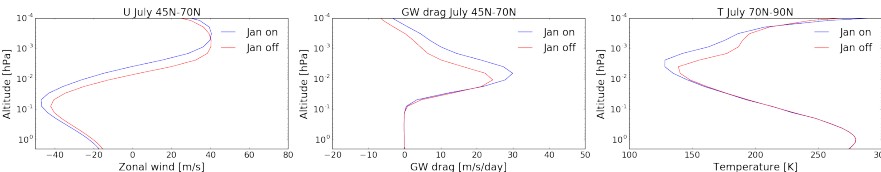

Fig. 3. Interhemispheric coupling in July, illustrated by the zonal wind and the GW drag between 45° and 70° N and the temperature between 70° and 90°N.




The blue lines show the control run and the red lines show the run without
GWs in the winter hemisphere.

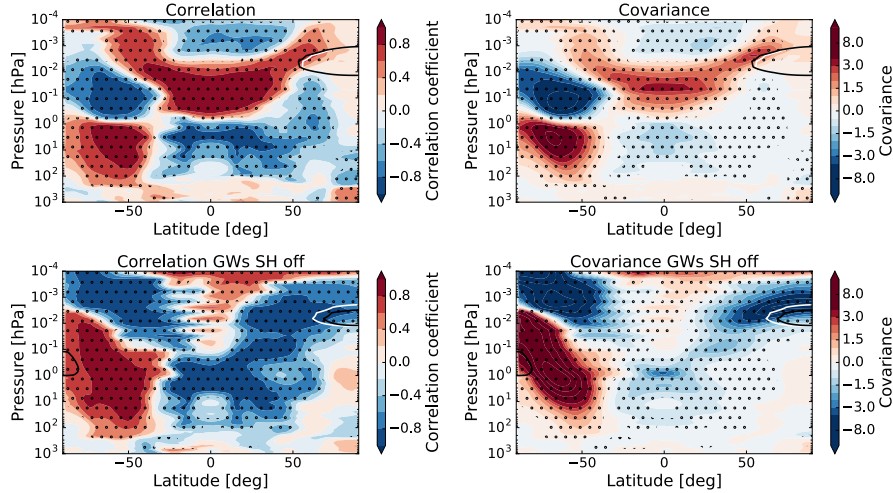


Fig. 4. The correlation (left) and covariance (right) between the temperature in
the winter stratosphere (1-10 hPa, 60°S-40°S) and the temperatures in the
rest of the atmosphere in July for the control run (first row) and run without
GWs in the winter hemisphere (bottom row). The dotted areas are regions
where the correlation has a p-value < 0.05. The black and the blue 150 K-
contour indicate the polar mesopause region.

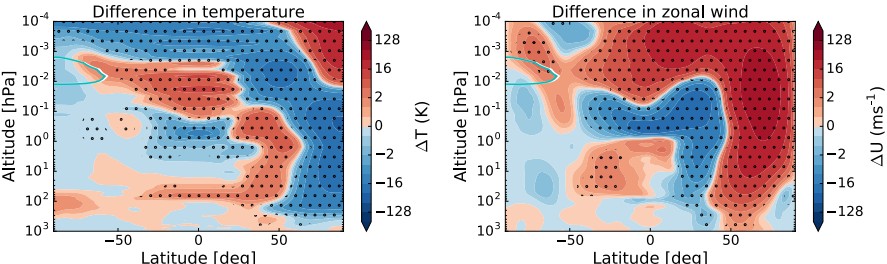

Fig. 5. Same as Figure 1, but for January.





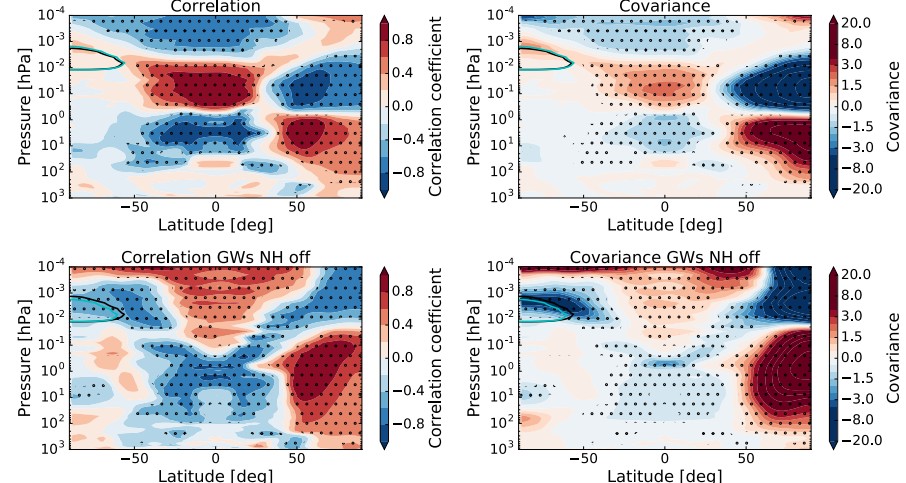


Fig. 6. The correlation (left) and covariance (right) between the temperature in
the winter stratosphere (1-10 hPa, 40°N-60°N) and the temperatures in the
rest of the atmosphere in January for the control run (first row) and run without
GWs in the winter hemisphere (bottom row). The black and the blue 150 K-
contour indicate the polar mesopause region. The dotted areas are regions
where the correlation has a p-value < 0.05.

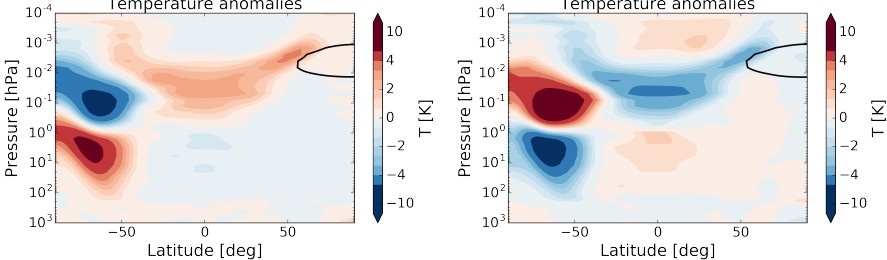

Fig. 7. The July temperature anomalies for anomalously high (left) and low
(right) temperatures in the winter stratosphere (1-10 hPa, 60°S-40°S) for the
control run. The black 150 K-contour indicates the polar mesopause region

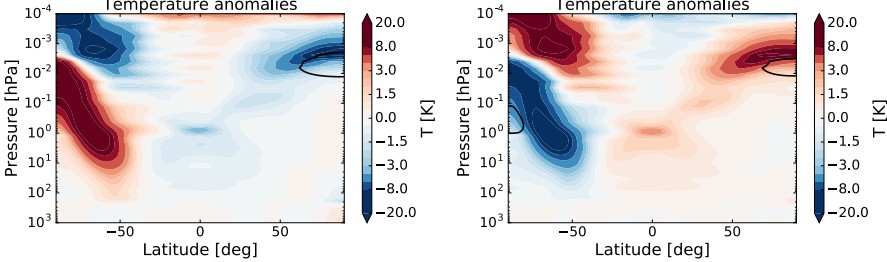




Fig. 8. The July temperature anomalies for anomalously high (left) and low
(right) temperatures in the winter stratosphere (1-10 hPa, 60°S-40°S) for the
run without GWs in the winter hemisphere. The black 150 K-contour indicates
the polar mesopause region.

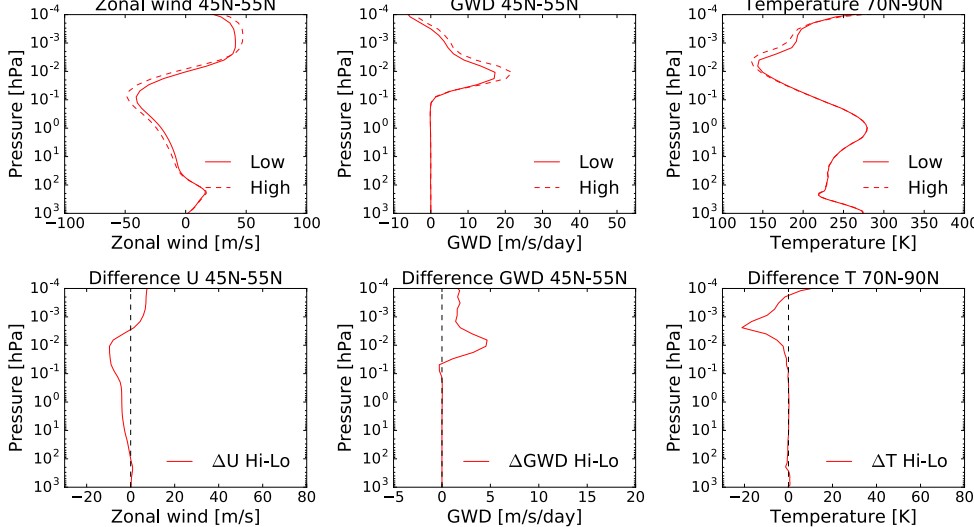


Fig. 9. The July zonal wind (left) and the GW drag (middle) between 45°-
55°N and the temperature (right) between 70-90°N for anomalously low and
high temperatures in the winter stratosphere (1-10 hPa, 60°S - 40°S) (first
row) and the differences between them (second row), for the case where
there are no GWs in the winter hemisphere. The red continuous lines show
the results for anomalously low temperatures, the red dotted lines show the
results for the anomalously high temperatures.

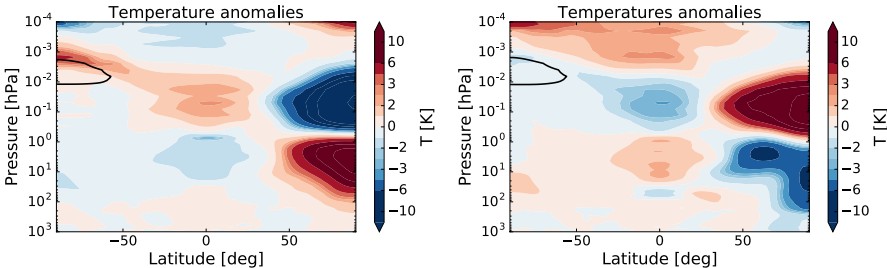


Fig. 10. The January temperature anomalies for anomalously high (left) and
low (right) temperatures in the winter stratosphere (1-10 hPa, 40°N-60°N) for





the control run. The black 150 K-contour indicates the polar mesopause
region.

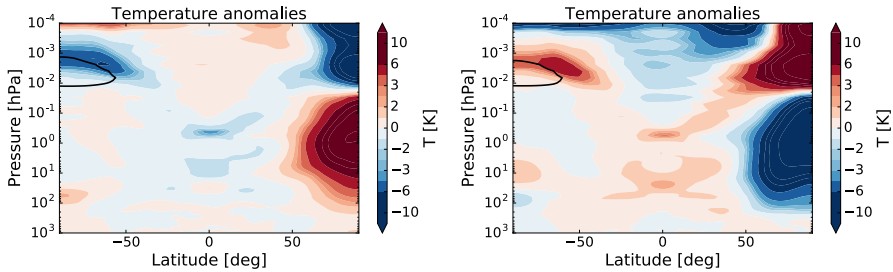


Fig. 11. The January temperature anomalies for anomalously high (left) and
low (right) temperatures in the winter stratosphere (1-10 hPa, 40°N-60°N) for
run without GWs in the winter hemisphere. The black 150 K-contour indicates
the polar mesopause region.

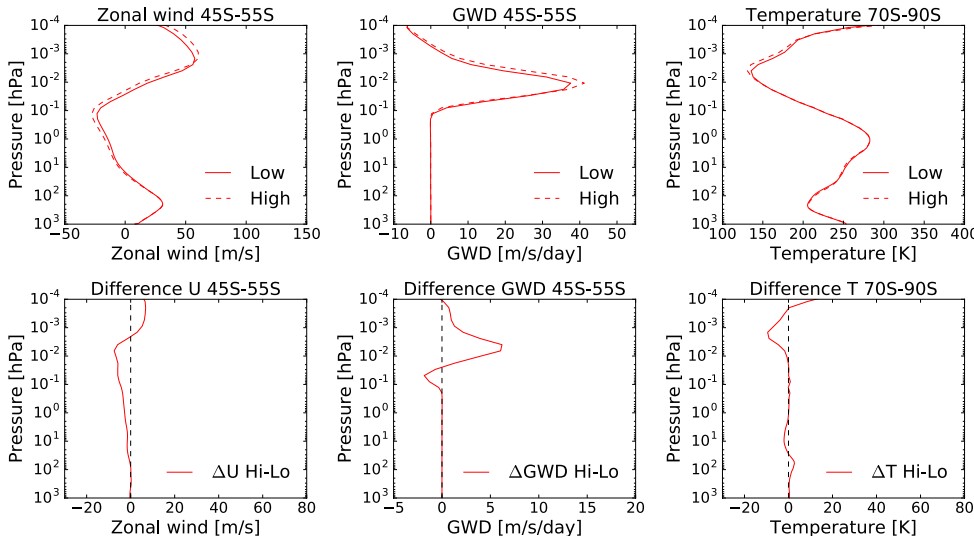


Fig. 12. The January zonal wind (left) and the GW drag (middle) between 45°-
55°S and the temperature (right) between 70°S-90°S for anomalously low and
high temperatures in the winter stratosphere (1-10 hPa, 40°N - 60°N) (first
row) and the differences between them (second row), for the case where
there are no GWs in the winter hemisphere. The red continuous lines show
the results for anomalously low temperatures, the red dotted lines show the
results for the anomalously high temperatures.