# Peer review of "The role of the winter residual circulation in the summer mesopause regions in"

_Atmospheric Chemistry and Physics, 2017_

## Referee Comment (RC1) · Anonymous Referee #2 · 22 Sep 2017

Review of "The role of the winter residual circulation in the summer mesopause regions in WACCM" by Maartje Kuilman and Bodil Karlsson (acp-2017-647).

The scientific question behind this paper is to what extent WACCM reflects the results of a KMCM study regarding the interhemispheric coupling mechanism published by Karlsson and Becker 2016 (hereafter: K+B16). The main focus lies on the interhemispheric coupling mechanism describing the impact of the winter stratosphere on the summer mesopause region. The authors are able to reproduce and reconfirm the results of K+B16 qualitatively to a large extent. However there are also differences in structure and magnitude of the effect that are not mentioned and discussed. In general

the paper has a very detailed introduction giving a good overview of the current status. The presentation of the results can be shortened since some figures include almost the same information. The idea of this study is solid and worth to publish. However a discussion and a valuation of how the WACCM results are comparable to that from KMCM, as promised in the abstract, are mostly missing Thus I recommend a publication after a major revision only.

Major comments: Line 75-82: The purpose of this paragraph is not clear.

Line 121: In this context is the anomalous cooling of the summer mesopause a real cooling or a shift in altitude of the summer mesopause?

Line 124-137: I think this paragraph is more suitable for the discussion part. However you argue that the QTDW is an additional mechanism without showing it nor discussing it later in the paper. Please remove this sentence and put this fundamental discussion in the discussion part later in the paper.

The introduction includes all that is needed and more but needs a new grouping in order to a better preparation of the reader for the results.

Line 265-267: What is the magnitude of the temperature increase and how is its relation to a radiation-only driven atmosphere?

The information one can get from figure 3 can also be get from figure one expect for the GW drag. I would suggest to add a plot of the difference in GW drag as a function of latitude and altitude in figure 1 and remove figure 3. This would also improve the understanding of the IHC mechanism for the reader. A valuation and discussion on how the WACCM results correspond to the KMCM results is missing not only for figure 1 and 3 but in general. A comparison of your figure 1 and figure 3 in K+B16 shows differences in magnitude and structure even though they qualitatively correspond to each other.

Figure 2 shows the difference in water vapor and ice mass resulting from the GWs.

The effect of the IHC on the NLC concurrency is interesting but the results are neither discussed nor brought in relation to other studies. Additionally I think that a discussion on this topic disrupts the central idea of the paper at this position. I would suggest to either remove the ice mass topic from the paper or to put it at the end so that the central idea of the paper is not interrupted.

Figure 4 shows the covariance of the control run and the run without GW in the SH for July. A critical comparison of these results with those of K+B16 (their figure 6) shows again a qualitatively agreement but differences in magnitude and also in structure. These differences should be mentioned and discussed.

Similar to figure 1, please insert the difference in GW drag in figure 5. Again a discussion and comparison of your results with those of K+B16 is missing. This is particularly important in the case of January since there are much larger differences between the results of WACCM and KMCM as it is the case for July. The same applies to figure 6.

In line 333-334 you hypothesized that the IHC less affects the SH summer. However, the magnitude of the IHC effect in the SH summer is weaker since it is more disturbed in the NH winter by planetary waves.

Line 361: Please describe shortly how a weak and strong BDC is defined here.

In section 3.1 the introductory text gives the impression that the effect of the summer stratosphere on the summer mesosphere is studied in the following. However, the descriptions of the figures 7 and 8 for July and figures 10 and 11 for January mostly replicate the results regarding the IHC shown in figure 4 and 6 and do not give a further insight into the effect of the summer stratosphere on the summer mesosphere. Additionally, the information taken from figures 7, 8, 10 and 11 can be obtained from figure 4 and 6 and therefore are redundant. I would like to see the results when you correlate the summer stratosphere with the rest of the atmosphere similar to your figure 4 and 6. Furthermore a discussion of this topic is missing and should be included.

The information from figure 9 and 12 can be obtained from figure 1 and 5 respectively and therefore are also redundant. However, a light discussion on the effect of the summer stratosphere on the summer mesosphere can be found in line 405-411 and 446-449 but none of the suggestions are shown or proven and are not compared to other studies.

Minor comments: o Line 34: . . .(e.g., Fritts and Alexander, 2003) o Line 59: . . . reversed with a cooling (warming) on top of the stratospheric warming (cooling) in the polar mesosphere -> your explanation is more clear without this o Line 51-62: You start the description of the IHC mechanism here and interrupt it for 40 lines. Especially for people without in depth knowledge of the IHC mechanism it is hard to follow you. It is better to describe the IHC mechanism in one go. o Line 121: . . ., with an anomalous cooling . . . o Line 144: please insert: . . .lower breaking GWs in the summer hemisphere and a warmer. . . o Line 161-171: The magnitude of the IHC effect is weaker in the SH summer mesopause than in the NH summer mesopause and not the impact. o Line 256: Please add: . . .parameterized GWs in the winter hemisphere. o Line 268-270: Please insert a reference.

---

## Referee Comment (RC2) · Anonymous Referee #1 · 9 Oct 2017

Review of manuscript acp-2017-647: "The role of the winter residual circulation in the summer mesopause regions in WACCM", by Maartje S. Kuilman and Bodil Karlsson.

This manuscript revisits the mesospheric Interhemispheric Coupling (IHC) contribution to control temperature in the summer mesopause, using the comprehensive climate model CESM/WACCM. The main result is that this model is able to reproduce the mechanism as shown by Karlsson and Becker (2016 J Clim, KB16) with the KMCM model . The manuscript is well written and structured, but the new scientific insights it offers are not clear. Regarding this, I have one general concern, and some specific comments, that the authors could address before meriting publication:

[Figure]

1) What is the motivation for trying to reproduce KB16 results with WACCM? Are there processes included in WACCM and not in KMCM that justify the study? It is relevant that Figs. 1 to 6 are basically the same figures as those in KB16, but with WACCM instead of KMCM. The authors could offer a detailed comparison between the two models, because those figures present some differences that are not highlighted in the text. For example, it would seem that the correlation is very weak in the NH summer polar mesopause in WACCM (Fig. 4 top left), but quite significant in KMCM (Fig. 8A in KB16).

Specific comments:

2) It would be interesting to include a discussion fn the effects of turning off the GWD on the Brewer-Dobson circulation (BDC) itself. In the experiments where the GWD in the winter hemisphere is turned off, does the amplitude of the planetary waves change? In other words, say the GWD represents 80% of the total wave forcing in the winter mesosphere; is w* 80% weaker in the experiments versus control? (i.e. does the EP flux divergence increase in the experiments, trying to compensate the missing GWD?)

3) Lines 302 and elsewhere. For the correlation, why is the SH temperature averaged over 40-60S, and not over polar latitudes (as the authors do in the NH)?

4) Figure 4 (and 6). If the point of these figures is to highlight the importance of the equatorial mesospheric temperatures on controlling the summer mesopause T, why not correlating the equatorial T (instead of extratropical T) with T elsewhere?

5) Lines 327-328. What is the NLC region? Is it the region bounded by the contour? If so, it is hard to see any response in temperature there.

6) Lines 358. It seems not quite conventional to use T in the extratropics as a proxy for the strength of the BDC, when the model provides with all the variables needed to calculate it. Please comment on this choice.

7) I wonder how necessary are Figures 7, 8, 10 and 11; they seem to provide the same

piece of information as Figures 4 and 6. I believe similar conclusions can be reached with the latter. Also, in lines 350-356 the authors decide to focus the discussion on the NH summer in July because of the stronger influence of the SH winter on the NH summer than vice versa. However, several paragraphs are devoted to this weaker connection between the NH winter on the SH summer. I recommend suppressing 412-450 (and the corresponding figures) for the sake of concision.

8) Section 3.1. I have some trouble trying to understand the objective of this section. Why is it interesting to discuss the role of the summer stratosphere on the summer mesospheric T in situations that are far from being realistic? Perhaps more interesting would be to perform an additional experiment in which the summer GWD is turned off. This way you can compare the importance of the summer BDC versus the IHC on the mesospheric T, and would definitely add new information from that given in KB16.

Technical comments:

- Figures: It is hard to see the dots that signal the statistical significance, and it is also quite difficult to assign a color to a value (in the colored figures). Perhaps adding black contours helps.

- Line 283: At the same "time"?

---

## Author Comment (AC1) · 19 Dec 2017

First of all, we would like to thank the reviewer for their constructive criticism, and time spent to analyze our manuscript. We are grateful for the valuable suggestions provided. The response to the comments and the new version of the manuscript are provided in the supplement.

Please also note the supplement to this comment:
https://www.atmos-chem-phys-discuss.net/acp-2017-647/acp-2017-647-AC1-supplement.zip

---

## Author Response (AR1)

This manuscript revisits the mesospheric Interhemispheric Coupling (IHC) contribution to control temperature in the summer mesopause, using the comprehensive climate model CESM/WACCM. The main result is that this model is able to reproduce the mechanism as shown by Karlsson and Becker (2016 J Clim, KB16) with the KMCM model. The manuscript is well written and structured, but the new scientific insights it offers are not clear. Regarding this, I have one general concern, and some specific comments, that the authors could address before meriting publication:

First of all, we would like to thank the reviewer for their constructive criticism, and time spent to analyze our manuscript. We are grateful for the valuable suggestions provided. Responses to each of the comments are listed below:

1) What is the motivation for trying to reproduce KB16 results with WACCM? Are there processes included in WACCM and not in KMCM that justify the study? It is relevant that Figs. 1 to 6 are basically the same figures as those in KB16, but with WACCM instead of KMCM. The authors could offer a detailed comparison between the two models, because those figures present some differences that are not highlighted in the text. For example, it would seem that the correlation is very weak in the NH summer polar mesopause in WACCM (Fig. 4 top left), but quite significant in KMCM (Fig. 8A in KB16).

WACCM is in some aspects a more comprehensive model than KMCM. E.g. a major difference is that WACCM contains interactive chemistry in the middle atmosphere, while KMCM does not. WACCM also uses a different parameterization for non-orographic GWs than KMCM. KMCM uses a simplified dynamical core and convection scheme as compared to WACCM. Moreover, the WACCM model is well-established within the community: this study confirms the results of the less known - yet advanced and high-performing - KMCM. Confirming that the responses are the same in a variety of models simply serves to strengthen the validity and robustness of our findings. We emphasize this on lines 161 – 169.

Please, note that Figure 8a in KB16 is from 8 years of MLS data (2005 – 2012) and not from the KMCM model. Figure 8e is showing the correlation from a 30-year run of the CMAM30 (which is as comprehensive as the WACCM) and as can be seen, the correlation coefficients have decreased considerably although they are significant and the structure is robust. If comparing Figure 8e to previous Figure 4a (now Figure 2 upper left), the correlation coefficients are similar. However, the responses differ in altitude and in latitudinal extent. We now point these differences out in the text: lines 312-324.

Specific comments:

2) It would be interesting to include a discussion of the effects of turning off
the GWD on the Brewer-Dobson circulation (BDC) itself. In the experiments
where the GWD in the winter hemisphere is turned off, does the amplitude of
the planetary waves change? In other words, say the GWD represents 80% of
the total wave forcing in the winter mesosphere; is w* 80% weaker in the
experiments versus control? (i.e. does the EP flux divergence increase in the
experiments, trying to compensate the missing GWD?)

This is for sure an intriguing question. We speculate that as the winter GWs
are removed, the eastward zonal flow will not be reversing into westward flow
in the mesosphere. Hence, the PWs could potentially propagate further up in
the stratosphere before reaching their critical levels (?). In such scenario, the
PW drag on the zonal flow would be distributed over a larger altitude range,
thus (since the drag is not so concentrated in a specific height region) the PW
would have a less dramatic impact on the zonal wind. The zonal flow
(attached below), particularly in the NH winter, is somehow confirming that.
We also note that in Figure 1, there is a significant warming signal in the
equatorial stratosphere indicating a weaker BD-circulation (which would agree
with less PW drag/GW drag). Moreover, when we composite into high (and
low) PW activity in the winter stratosphere, the warming (cooling) anomaly
form the enhanced (reduced) BD-circulation extends into the mesosphere
(see e.g. figure 2, left, bottom row, where we would otherwise have a cooling
(warming) as a response of the GW drag (see figure 2, left, top row). We
won't go into further details about what happens to the PWs in the winter
stratosphere/mesosphere this study.

[Figure]

Zonal wind profiles July for the latitude band 60°S-80°S (left) and January for the latitude
band 60°N-80°N (right).

For your information, I do show the Eliassen-Palm flux and Eliassen-Palm flux
divergence. The EP flux divergence does indeed increase in the winter
stratosphere, if there are no GWs in the winter hemisphere, suggesting that
the amplitude of the PWs changes. We don't investigate this further for this
study.

[Figure]

Eliassen-Palm flux for July for the control case (left) and the case where there are no GWs in
the NH (right).

[Figure]

[Figure]

Eliassen-Palm flux divergence July for the control case (above, left) and the case where there
are no GWs in the NH (above, right) and difference between them (below).

3) Lines 302 and elsewhere. For the correlation, why is the SH temperature
averaged over 40-60S, and not over polar latitudes (as the authors do in the
NH)?

This is because in the SH, the PW forcing is weak so that the residual flow
does not reach the highest latitudes (see Kuroda and Kodera, 2001; their
figure 4). This is now clarified on lines 300-304.

l. 302-306. *"The latitude and altitude ranges chosen for July is the region
where the SH winter stratosphere variability is best captured (see Karlsson
and Becker, 2016; their figure 9). This is related to the relatively weak PW
forcing in the SH – the BDC is not reaching all the way to the polar region
(Kuroda and Kodera, 2001)."*

4) Figure 4 (and 6). If the point of these figures is to highlight the importance
of the equatorial mesospheric temperatures on controlling the summer
mesopause T, why not correlating the equatorial T (instead of extratropical T)
with T elsewhere?

This can also be done and as shown below first for July then January, the
results look similar as shown below. The idea was to start from the
strong/weak BDC and then explain the mechanism behind the temperature
change in the equatorial mesosphere and the effect on the summer
mesosphere.

[Figure]

The temperature anomaly field for July taking the equatorial mesosphere as a proxy (20°S –
20°N, 0.13-0.01 hPa) for the GWs on.

[Figure]

The temperature anomaly field for July taking the equatorial mesosphere as a proxy (20°S-
20°N, 0.13-0.01 hPa) for the GWs in the SH off.

[Figure]

Correlations and covariance with the equatorial mesosphere (20°S-20°N, 0.13-0.01 hPa) in
July.

[Figure]

The temperature anomaly field for January taking the equatorial mesosphere as a proxy
(20°S – 20°N, 0.13-0.01 hPa) for the GWs on.

[Figure]

The temperature anomaly field for January taking the equatorial mesosphere as a proxy
(20°S-20°N, 0.13-0.01 hPa) for the GWs in the SH off.

[Figure]

Correlations and covariance with the equatorial mesosphere (20°S-20°N, 0.13-0.01 hPa) in January.

5) Lines 327-328. What is the NLC region? Is it the region bounded by the contour? If so, it is hard to see any response in temperature there.

No that is true, there is no clear increase in temperature in this region. There is a small positive correlation in this region as can be seen in Fig. 4. However, this change is not statistically significant, this is something we can understand as explained in the introduction.

6) Lines 358. It seems not quite conventional to use T in the extratropics as a proxy for the strength of the BDC, when the model provides with all the variables needed to calculate it. Please comment on this choice.

The EP-flux divergence is not given as an output in WACCM. Since it is evident from Karlsson et al. 2007 and 2009 that the winter stratospheric temperature is an excellent proxy for the PW activity, we decided to use what was available. However, we ended up calculating the EP-flux divergence anyway (see above), but only for the 30-year mean. We hope that the reviewer is satisfied with our motivation for using the temperatures for compositing between high and low PW activity instead of the EP-flux because it was quite time consuming to calculate the EP-flux divergence and we need to remake that computation for all the 30 years. To assure that the results are very similar, we can carry out the EP-flux divergence calculations for each and every year, but only if the reviewer find it necessary.

7) I wonder how necessary are Figures 7, 8, 10 and 11; they seem to provide the same piece of information as Figures 4 and 6. I believe similar conclusions can be reached with the latter. Also, in lines 350-356 the authors decide to focus the discussion on the NH summer in July because of the stronger influence of the SH winter on the NH summer than vice versa. However, several paragraphs are devoted to this weaker connection between the NH winter on the SH summer. I recommend suppressing 412- 450 (and
the corresponding figures) for the sake of concision.

We agree that the information from the mentioned figures can be derived from
Fig. 1 and the new Fig. 2.  The section about the influence of the summer
stratosphere has now been made shorter and more to the point.

8) Section 3.1. I have some trouble trying to understand the objective of this
section. Why is it interesting to discuss the role of the summer stratosphere
on the summer mesospheric T in situations that are far from being realistic?

The section on the summer stratosphere has been rewritten. We hope the
introduction to this section now gives a clearer picture on what is done.
l.386-391. *"The BDC is modifying in the summer stratospheric meridional*
*temperature gradient. Hence, filtering effects taking place below the*
*mesosphere may seem like an additional - or alternative – mechanism to the*
*response observed in the summer mesopause. In this section, we will discuss*
*why this cannot be the case. We focus again mostly on the NH summer polar*
*mesosphere region."*

Perhaps more interesting would be to perform an additional experiment in
which the summer GWD is turned off. This way you can compare the
importance of the summer BDC versus the IHC on the mesospheric T, and
would definitely add new information from that given in KB16.

This simulations have been done already, as they come automatically when
one runs the whole year without the GWs in the SH or NH. The problem with
looking at these data is that without the GWs in the summer hemisphere,
there is no summer mesopause region at all. The summer GWs are crucial for
making the summer mesopause cold: the winter flow is only modulating where
the summer GWs break. For further information, see the study by Körnich and
Becker, 2011: they show that the IHC signal is not communicated to the
summer mesosphere when the summer GWs are absent.

[Figure]

Temperature July (left) and temperature July when the GWs in the NH off (right)

Technical comments:

Figures: It is hard to see the dots that signal the statistical significance, and it is also quite difficult to assign a color to a value (in the colored figures). Perhaps adding black contours helps.

Technical comments:

Figures: It is hard to see the dots that signal the statistical significance, and it
is also quite difficult to assign a color to a value (in the colored figures).
Perhaps adding black contours helps.

I agree that it was hard to see, the figures are now quite small and adding
more contours makes the figures a bit chaotic. Instead what is done now, is
shading the areas in which the confidence level of 95% is not reached.

- Line 283: At the same "time"?

Yes, this was what was meant, this section has now been removed though.

*Interactive comment on* "The role of the winter residual circulation in the
summer mesopause regions in WACCM" *by* Maartje Sanne Kuilman and
Bodil Karlsson

**Anonymous Referee #2**

The scientific question behind this paper is to what extent WACCM reflects
the results of a KMCM study regarding the interhemispheric coupling
mechanism published by Karlsson and Becker 2016 (hereafter: K+B16). The
main focus lies on the interhemispheric coupling mechanism describing the
impact of the winter stratosphere on the summer mesopause region. The
authors are able to reproduce and reconfirm the results of K+B16 qualitatively
to a large extent. However there are also differences in structure and
magnitude of the effect that are not mentioned and discussed. In general the
paper has a very detailed introduction giving a good overview of the current
status. The presentation of the results can be shortened since some figures
include almost the same information. The idea of this study is solid and worth
to publish. However a discussion and a valuation of how the WACCM results
are comparable to that from KMCM, as promised in the abstract, are mostly
missing. Thus I recommend a publication after a major revision only.

First of all, we would like to thank the reviewer for their constructive criticism,
and time spent to analyze our manuscript. We are grateful for the valuable
suggestions provided. Responses to each of the comments are listed below:

Major comments: Line 75-82: The purpose of this paragraph is not clear.

The text is now rewritten in order to make clear which purpose these
paragraph serves.

l.64-73:*"These anomalies are responses to different wave forcing in the winter*
*hemisphere. To understand how these anomalies come about we have to*
*understand the interhemispheric coupling mechanism. The mechanism, as*
*discussed here, is for the case of a stronger winter residual circulation, but*
*works the same for a weakening of this circulation (Karlsson et al., 2009). A*
*stronger planetary wave forcing in the winter stratosphere yields a stronger*
*stratospheric Brewer-Dobson circulation (BDC). This anomalously strong flow*
*yields an anomalously cold stratospheric tropical region and a warm*
*stratospheric winter pole, due to the downward control principle (Haynes et al.*
*1991)."*

Line 121: In this context is the anomalous cooling of the summer mesopause
a real cooling or a shift in altitude of the summer mesopause?

If the gravity waves break at a higher altitude, the summer mesopause will be
colder. This is a real cooling: lower temperatures are reached in the
mesopause, as can be seen in the figure below.

[Figure]

Temperature July for the control case (left) and the case, for which there are no GWs in the
SH (right). The blue contour indicates the region where the temperature is below 150 K.

Line 124-137: I think this paragraph is more suitable for the discussion part.
However you argue that the QTDW is an additional mechanism without
showing it nor discussing it later in the paper. Please remove this sentence
and put this fundamental discussion in the discussion part later in the paper.

This part was put in the introduction because the debated status of IHC
mechanism is an additional motivation for this study. However, we understand
the objections the reviewer has against this section, indeed this is not further
studied in this paper and has now been removed.

The introduction includes all that is needed and more but needs a new
grouping in order to a better preparation of the reader for the results.

The introduction has now been reordered and there is a new section (l.161-
190) explaining what will be done in this study, we hope it is now clearer for
the readers what is going to be discussed.

Line 265-267: What is the magnitude of the temperature increase and how is
its relation to a radiation-only driven atmosphere?

The temperature increase in the NLC region, which I have now defined to be
between 61°N - 90°N and 0.01 - 0.002 hPa, is approximately 16 degrees.

In a radiation only atmosphere the temperature in the NH NLC region is about
210-220 K. Without GWs in the winter hemisphere, there is still a mesopause
region, as can be seen in temperature fields for July as shown as response to
an earlier comment.

The information one can get from figure 3 can also be get from figure one
expect for the GW drag. I would suggest to add a plot of the difference in GW
drag as a function of latitude and altitude in figure 1 and remove figure 3. This
would also improve the understanding of the IHC mechanism for the reader. A
valuation and discussion on how the WACCM results correspond to the
KMCM results is missing not only for figure 1 and 3 but in general. A
comparison of your figure 1 and figure 3 in K+B16 shows differences in
magnitude and structure even though they qualitatively correspond to each
other.

A plot with the changes in the GWD is added to figure 1. Figure 3 is now
removed. A section discussing the differences has now been added. However,
the point of this study is not so much to explain in detail how the differences in
responses between KMCM and WACCM come about, but rather to reconfirm
that in the absence of winter gravity waves, there is a warming of the summer
mesopause region and to strengthen the evidence for the interhemispheric
coupling mechanism, with the equatorial mesosphere region as crucial region
of importance.

l. 275-290. *"When we compare our results with the results in Karlsson and*
*Becker (2016, their figure 3), we observe there are some quantitative*
*discrepancies in the structure of the responses. For example, Karlsson and*
*Becker (2016) found that removing the winter GWs resulted in a warming of*
*the mesosphere globally, although the response was strongest in the polar*
*mesopause region. They attributed that the warming over the equatorial and*
*winter mesosphere to the effect that GWs have on tides: when GWs are*
*absent, the tidal response is enhanced. The same behavior is not found in*
*WACCM - in fact, the equatorial upper mesosphere is anomalously cooler*
*when the GWs are removed. These differences could perhaps be explained*
*by for example the different gravity wave parameterization of non-orographic*
*GWs, the different dynamical cores between the models and the presence of*
*interactive chemistry in the middle atmosphere in WACCM. However, the*
*qualitative response of the temperature and zonal wind change due to turning*
*of the GWs in the SH corresponds well with the results from the KMCM as*
*well as with our hypothesis."*

Figure 2 shows the difference in water vapor and ice mass resulting from the
GWs. The effect of the IHC on the NLC concurrency is interesting but the
results are neither discussed nor brought in relation to other studies.
Additionally I think that a discussion on this topic disrupts the central idea of
the paper at this position. I would suggest to either remove the ice mass topic
from the paper or to put it at the end so that the central idea of the paper is
not interrupted.
We understand the objections the reviewer has to this section. We agree that
this disrupting the main point of the paper. This part has now been removed.
Figure 4 shows the covariance of the control run and the run without GW in
the SH for July. A critical comparison of these results with those of K+B16
(their figure 6) shows again a qualitatively agreement but differences in
magnitude and also in structure. These differences should be mentioned and
discussed.
A comparison with the results of K+B16 has now been added. However, as
stated before the point of this study is not so much to explain in detail how the
differences in responses between KMCM and WACCM come about.
l. 312-324. *"Comparing the results show in Figure 2 (upper left) to Figure 8e in*
*Karlsson and Becker (2016), it can be seen that the correlation coefficients*
*are of similar magnitudes, but the spatial responses differ in altitude and in*

*latitudinal extent: whereas the correlation signal is significant in the CMAM30*
*July high latitude summer mesopause, the WACCM July response reaches*
*only the lowermost latitudes (about 50°N in latitude).*
*If the GWs are removed in the winter hemisphere, the temperature in the*
*summer mesopause region anti-correlates with the temperature in the winter*
*stratosphere. Also, the temperature in the equatorial mesosphere does no*
*longer correlate and co-vary significantly with the temperature in the winter*
*hemisphere, in agreement with the results of Karlsson and Becker (2016)."*

Similar to figure 1, please insert the difference in GW drag in figure 5. Again a
discussion and comparison of your results with those of K+B16 is missing.
This is particularly important in the case of January since there are much
larger differences between the results of WACCM and KMCM as it is the case
for July. The same applies to figure 6.

The GW drag has now been inserted in figure 5. A comparison with the
results of K+B16 has now been added.

l. 353- 361. *"Comparison between the responses found using WACCM with*
*those found with KMCM (Karlsson and Becker, 2016, their Fig. 3), shows that*
*the temperature change is larger and extends all the way to the summer pole*
*in KMCM, while this is not the case in WACCM. Moreover, the change in*
*temperature in this region is not statically significant in WACCM. The*
*differences in temperature and zonal wind responses are larger in January*
*than in July when comparing the results of WACCM with that of KMCM.*
*Nevertheless, the qualitative structure of the temperature and zonal wind*
*change due to turning of the winter GWs corresponds convincingly well."*

In line 333-334 you hypothesized that the IHC less affects the SH summer.
However, the magnitude of the IHC effect in the SH summer is weaker since it
is more disturbed in the NH winter by planetary waves.

It is right that there are more planetary waves in the NH winter.  This means
that there is a stronger Brewer-Dobson circulation in NH winter – thus a
weaker zonal flow. This allows for the upward propagation of more GWs with
an eastward phase speed, which reduces the westward GW drag. This results
in a reduction in the strength of the winter-side mesospheric residual
circulation, which causes an anomalous warming of the equatorial
mesosphere as compared to the case where there would be less planetary
waves in the winter hemisphere. This explains why the equatorial mesosphere
is substantially colder in July than in January.

A warmer equatorial mesosphere leads to a positive temperature anomaly in
the summer mesopause.  Since the NH winter stratosphere zonal flow
oscillates between being weak and strong, the equatorial mesosphere is
modified continuously: it varies between being cooled and warm, so – if
thinking about it in a more 'climatological sense' – the effect of IHC is not
going to be as strong as for the SH winter, when the eqatorial region is
constantly cooled by the strong residual flow. Taking away the GWs in the NH

winter will have a smaller effect on the SH summer mesopause than taking
away the GWs in the SH winter on the NH summer mesopause, as there is
already less GW drag in the NH winter as compared to the SH winter.

Hence, the interhemispheric coupling mechanism gives a plausible
explanation to why the July summer mesosphere region is considerably
colder than the one in January. This is now clarified on lines 345-352.

Line 361: Please describe shortly how a weak and strong BDC is defined here.

This section has been rewritten:
l.393-407. *"In Fig. 1, it is seen that if there are GWs in the SH winter*
*hemisphere the temperature in the winter stratosphere is positively correlated*
*with the temperature in the NH summer polar mesosphere. This means that*
*for a stronger Brewer-Dobson circulation (BDC) and the resulting anomalously*
*warm (cold) temperatures in the stratosphere at 40°- 60°S, there will be also*
*an anomalously warm (cold) temperature in the summer polar mesosphere.*
*A strong or weak BDC results in a temperature change in the equatorial*
*mesosphere, which changes the meridional temperature gradient in the*
*summer mesosphere. As a result of the change in strength of the BDC, there*
*is a change in the meridional temperature gradient as well, however, this*
*gradient will have an opposite sign, as can be seen from Fig 1."*

In section 3.1 the introductory text gives the impression that the effect of the
summer stratosphere on the summer mesosphere is studied in the following.
However, the descriptions of the figures 7 and 8 for July and figures 10 and
11 for January mostly replicate the results regarding the IHC shown in figure 4
and 6 and do not give a further insight into the effect of the summer
stratosphere on the summer mesosphere. Additionally, the information taken
from figures 7, 8, 10 and 11 can be obtained from figure 4 and 6 and therefore
are redundant.
Section 3.1 has been rewritten. The introduction explains the purpose now
hopefully more clear:
l.386-391. *"The BDC is modifying in the summer stratospheric meridional*
*temperature gradient. Hence, filtering effects taking place below the*
*mesosphere may seem like an additional - or alternative – mechanism to the*
*response observed in the summer mesopause. In this section, we will discuss*
*why this cannot be the case. We focus again mostly on the NH summer polar*
*mesosphere region."*
I would like to see the results when you correlate the summer stratosphere
with the rest of the atmosphere similar to your figure 4 and 6. Furthermore a
discussion of this topic is missing and should be included.
Below we include figures of correlations and composites studies that start out
in the summer stratosphere. As can be seen, if there is variability in the
summer stratosphere, this will indeed influence the summer mesopause. E.g.

if we had a large variability in the year-to-year ozone heating, this would
probably influence the summer mesopause via GW filtering. It is however not
so easy to sort out what drives variability in the summer stratosphere. From
the correlation plot, the IHC pattern jumps out even though the correlation
point is set in the summer stratosphere (which by the way varies very little
from one year to another, as confirmed by the composite studies below
(anomalous T-fields).
Hence, we argue that the variability (globally) is driven by PWs in the winter
hemisphere: via the BDC the summer stratospheric temperatures are slightly
modified and via the winter mesospheric flow, the summer mesopause
temperatures are affected. Our point is to show that the temperature response
to the variability in the summer mesopause really goes via the equatorial
mesosphere, and not via the summer stratosphere. We can verify this by
removing the GWs in the winter and show that the mesospheric response of
the variability in the summer stratosphere has the opposite sign (see figure 2).

[Figure]

Correlations and covariance with the summer stratosphere (52°N-90°N, 1-100 hPa) in July.

[Figure]

The temperature anomaly field for July taking the summer stratosphere (52°N-90°N, 1-100
hPa) as a proxy for the control case.

[Figure]

The temperature anomaly field for July taking the summer stratosphere (52°N-90°N, 1-100
hPa) as a proxy for the GWs in the SH off.

[Figure]

Correlations and covariance with the summer stratosphere (52°S-90°S, 1-100 hPa) in
January.

[Figure]

The temperature anomaly field for January taking the summer stratosphere (52°S-90°S, 1-
100 hPa) as a proxy for the control case.

[Figure]

The temperature anomaly field for July taking the summer stratosphere (52°N-90°N, 1-100
hPa) as a proxy for the GWs in the SH off.

The information from figure 9 and 12 can be obtained from figure 1 and 5
respectively and therefore are also redundant. However, a light discussion on
the effect of the summer stratosphere on the summer mesosphere can be
found in line 405-411 and 446-449 but none of the suggestions are shown or
proven and are not compared to other studies.

It may be true that it is possible to derive the information from Fig. 9 and 12
from Fig. 1 and 2, but it is not that easy to see. The profiles show what the
point we want to make in this section. Comparing Fig. 9 and 12 also shows
that that even though the signal is weaker in the SH, the general pattern of in
the regions of interest are very similar.

Minor comments:

Line 34: ...(e.g., Fritts and Alexander, 2003)

l.35. *(e.g., Fritts and Alexander, 2003).*

Line 59: ... reversed with a cooling (warming) on top of the stratospheric
warming (cooling) in the polar mesosphere -> your explanation is more clear
without this

I don't really understand what the reviewer means here. I stated that the IHC
pattern manifests itself as a quadruple structure in the temperature fields in
the winter hemisphere. In the sentence before this part I explain the
temperature anomalies in the stratosphere. Then I have temperature
anomalies in the mesosphere as well, otherwise it is not clear that there is a
quadrupole structure. I reformulated this part, I hope it is clearer now.

*l.55-62. "Its pattern consists of a quadruple structure in the winter hemisphere*
*with a warming (cooling) of the polar stratosphere and an associated cooling*
*(warming) in the equatorial stratosphere. In the mesosphere, these anomalies*
*are reversed: there is a cooling (warming) in the polar mesosphere, and an*
*associated warming (cooling) in the equatorial region. The mesospheric*
*warming (cooling) in the tropical region extends to the summer mesopause*
*(see e.g. Körnich and Becker, 2010)."*

Line 51-62: You start the description of the IHC mechanism here and interrupt it for 40 lines. Especially for people without in depth knowledge of the IHC mechanism it is hard to follow you. It is better to describe the IHC mechanism in one go.

The idea was to give first an introduction to the mechanism and give a quick qualitative discussion and then give a detailed discussion. But I agree it might be clearer if I change the structure. The text has been reordered.

Line 121: ..., with an anomalous cooling ...

This has been changed.

l. 111-115. *"In the case of an equatorial mesospheric cooling, the response is the opposite: the relative difference between the zonal flow and the phase speeds of the gravity waves increase to that they break at a slightly higher altitude, with an anomalous cooling of the summer mesopause as a result."*

Line 144: please insert: ... lower breaking GWs in the summer hemisphere and a warmer...

This has been inserted.

l. 129-133. *"Karlsson and Becker (2016) hypothesized that if the GW-driven winter residual circulation would not be present, the equatorial mesosphere would be warmer, which would lead to lower breaking levels of GWs in the summer hemisphere and a warmer summer mesosphere region."*

Line 161-171: The magnitude of the IHC effect is weaker in the SH summer mesopause than in the NH summer mesopause and not the impact.

l. 161-167, the text has been changed.

l.143-150. *"If – as hypothesized by Karlsson and Becker (2016) – the fundamental effect of the IHC is a cooling of the summer mesopauses, it would mean that the mechanism plays a more important role affecting the temperatures in the summer mesopause in the NH compared to that in the SH, since the weaker planetary wave activity in the SH results in an increased gravity wave drag and a strengthening of mesospheric poleward flow in the winter mesosphere. The equatorial mesosphere is adiabatically cooled more efficiently than when the winter mesospheric circulation is weak."*

For the part 167-171: I don't understand the objection the reviewer has against this formulation?

*Karlsson and Becker (2016) hypothesized that in the absence of the equator-to-pole flow in the SH winter, the summer mesopause in the NH would be considerably warmer.  Moreover, removing the mesospheric residual*

*circulation in the NH winter would not have as high impact on the SH summer*
*mesopause.*

Line 256: Please add: ...parameterized GWs in the winter hemisphere.

This has been added.

I. 229-231. *"In the perturbation runs, the equator-to-pole flow is removed by*
*turning off the parameterized gravity waves in the winter hemisphere."*

Line 268-270: Please insert a reference.

The reference has been added.

[revised manuscript text omitted]

---

## Author Response (AR2)

**Important changes in the manuscript**

Following the comments of the reviewers, quite some changes in the manuscript have been made.

We have made the relevance of this study clearer and changed the presentation of the results.

We now show the effect of taking away the gravity waves on the EP flux divergence and residual circulation in Figure 1 and 2.

**Suggestions for revision or reasons for rejection (will be published if the paper is accepted for final publication)**

Review of manuscript acp-2017-647: "The role of the winter residual circulation in the summer mesopause regions in WACCM", by Maartje S. Kuilman and Bodil Karlsson.

The authors have substantially reduced the length of the manuscript, and modified part of the text, in response to my comments. I think the paper goes more to the point, and thus has improved. However, I am not fully convinced about the novelty of the results; the response of the authors to this concern has been a bit vague.
I am somewhat torn on what to recommend for this paper. I think there is some interest in these results, but I believe the paper needs some new figures and a clarification of certain aspects of the presentation of the results, before it is ready for publication.

First of all, we would like to thank the reviewer for constructive criticism, and time spent to analyze our manuscript again. We are grateful for the valuable suggestions provided. Responses to each of the comments are listed below.

Major concerns:

- What is the motivation for trying to reproduce KB16 results with WACCM? Are there processes included in WACCM and not in KMCM that justify the study?
I am not entirely satisfied with the response given to this question. WACCM includes a chemistry module in the middle atmosphere, and has different GW parameterizations and dynamical cores than those in KMCM. But this is generally true for any pair of general circulation models. What have we learned from this paper that we did not know from the previous papers, particularly KB16?
With this comment I would like to encourage the authors to find an attractive way to present the results and highlight their relevance. The way the paper is motivated in the Introduction section (and the way the results are summarized in the Conclusions section), gives me the impression that the paper is an exercise of reproducibility of previous results – which is always good news, but perhaps not enough for an article to be published in ACP.

As the reviewer states, WACCM contains interactive chemistry and has a more sophisticated dynamical core, so in that sense it does contain processes that are not in KMCM.

An important complement to the study of KB16 is that we investigate also the role of the *stratosphere* in shaping the conditions of the summer polar. Using composite analyses, we show that in the absence of an anomalous summer mesospheric temperature gradient between the equator and the polar region, weak planetary wave forcing in the winter would lead to a warming of the summer polar mesosphere region instead of a cooling, and vice versa. This is opposing the temperature signal of the interhemispheric coupling in the mesosphere, in which a cold and calm winter stratosphere goes together with a cold summer mesopause.

We also show how the EP flux divergence and residual circulation are affected by removing the GWs in the winter hemisphere, which was not done in KB16.

- Figures 1 and 2. These figures show two things: 1) the IHC mechanism controls the mean T of the summer polar mesopause (Fig. 1); and 2) that T in the summer polar mesopause does not covary statistically significantly with T in the winter stratosphere (Fig. 2). The authors seem to use interchangeably "summer mesopause" and "summer polar mesopause" throughout the paper; but Fig. 2 shows that the summer mesopause away form the pole has statistically significant anomalies of T, while this is not the case for the polar region.

We agree that we used the terms 'summer mesopause' and 'summer polar mesopause' not very precise, this has now been changed.

It is true that the temperature anomalies in the summer polar mesosphere not in all the cases reaching a confidence level of 95% all the way to the poles. We comment on this in line 382-387:

*" This is in agreement with the results presented in Karlsson et al. (2009) although the WACCM temperature response does not reach statistical significance at a 95% level all the way to the polar region. This could be due to time lags between the response in the summer mesopause and the dynamic activity in the winter: Karlsson et al. (2009) found a lag between the winter and the summer hemisphere of up to 15 days. In the monthly-mean approach that we use for this study, lags in time are not accounted for."*

So what controls the summer polar mesopause T in a climatological (average) sense? IHC is clearly shown to play a role. It would also be nice to mention that the presence of GW in the summer hemisphere is a crucial factor, as shown in the response figures. Without them, the summer polar mesopause will be very high (or even absent).

It is atmospheric gravity waves that are responsible for the low temperatures in the summer polar mesosphere.
- Atmospheric gravity waves drive the circulation in the middle atmosphere. When they break, they deposit their momentum into the background flow, creating a drag on the zonal winds in the mesosphere, which establishes the pole-to-pole circulation This circulation drives the temperatures far away from the state of radiative balance, by adiabatically heating the winter mesopause and adiabatically cooling the summertime mesopause. The adiabatic cooling in the summer leads to temperatures sometimes lower than 130 K in the summer polar mesopause. (Introduction)

- "In a radiation-driven atmosphere the temperature in the summer polar mesosphere is about 210-220 K, which is much higher than the temperature both with and without the GWs in the SH." (Discussion of Fig.1)

As for the interannual variability, the IHC only seem to reach 50N, but not further north (line 376-383).

See earlier comment.

- I really think the paper would benefit from including plots of the EP flux divergence and the Tranformed Eulerian Mean velocities (either v* or w*). Those are more direct measures of the wave forcing and the BDC than stratospheric T, and they should not be difficult to compute from monthly mean output (WACCM does provide output of zonal mean flux diagnostics v'T', u'v' etc that speeds up the process). And this way I believe it would add value to the results as compared to KB16.

The fields of EP flux divergence and the TEM velocities are now included in figure 1 and 2 giving insight on how the EP flux divergence and the residual circulation velocities are changing due to the removal of GWs in the winter hemisphere.

Specific comments:

- Line 133. Analogically → Analogously?

This has been changed.

- Line 176. Simply WACCM since it has already been defined.

This has been changed.

- Line 272: degrees → K (Kelvin).

This has been changed.

- Line 280. They attributed that the warming → They attributed the warming.

This has been changed.

- Line 288: turning of → turning off

This has been changed.

- Lines 289-290. I may have misunderstood it, but if T in the equatorial mesosphere has a different sign in both models, there is no qualitative agreement.

I agree that this part wasn't explained clearly.

The temperature response in the equatorial mesosphere region of our interest doesn't have a different sign. Note that in our Fig. 1 we show the results for the run without the winter GWs minus the control run, whereas KB16 show them the other way around.

 The temperature response in the upper part of the mesosphere is different, however this region is not of interest for our discussion. The text has now been rewritten.

*"When we compare our results with the results in Karlsson and Becker (2016, their figure 3), we observe there are some quantitative discrepancies in the structure of the responses. For example, Karlsson and Becker (2016) found that removing the winter GWs resulted in a warming of the upper mesosphere globally, although the response was strongest in the polar mesopause region. They attributed the warming over the equatorial and winter mesosphere to the effect that GWs have on tides: when GWs are absent, the tidal response is enhanced. The same behavior is not found in WACCM - in fact, the equatorial upper mesosphere is anomalously cooler when the GWs are removed. These differences could perhaps be explained by for example the different gravity wave parameterization of non-orographic GWs, the different dynamical cores between the models and the presence of interactive chemistry in the middle atmosphere in WACCM.*

*However, the upper mesospheric response is not affecting the mechanism, we are discussing in this study. We don't consider the upper mesosphere region in the rest of the paper. The qualitative response of the temperature and zonal wind change in the stratosphere and lower parts of mesosphere due to turning of the GWs in the SH corresponds well with the results from the KMCM as well as with our hypothesis."*

- Line 343: less effect → weaker effect.

This has been changed.

- Lines 358. It seems not quite conventional to use T in the extratropics as a proxy for the strength of the BDC, when the model provides with all the variables needed to calculate it. Please comment on this choice.

We use the wintertime stratospheric temperature to look at the difference between the two cases. We could use the meridional component of the EP flux or the EP flux divergence, however there is always a delay in the wave forcing and the response in the temperature, therefore the patterns become more clear using the temperature. Note that the same is done in KB16.

- Line 386. I do not understand these sentences. Does it mean that the BDC modifies the summer stratospheric meridional gradient of T?

This is the case, this sentence has now been rewritten.

"The summer stratospheric meridional temperature gradient is affected by the strength of Brewer-Dobson circulation."

- Lines 395-398. But in Fig. 2 we see that the anomalies are not statistically significant.

The temperature anomalies show the pattern of IHC extending further northwards, but it is true that the temperature anomalies in the summer polar mesosphere not in all the cases reaching a confidence level of 95% all the way to the poles.

- Lines 468-471. "… it is the equatorial mesosphere that is governing the temperature in the summer mesopause regions...". I agree, but the equatorial T is ultimately driven by the BDC changes forced by the winter planetary wave forcing, according with your results.

This formulation has now been changed.

*"This confirms the idea that the net effect of the IHC mechanism, with the equatorial mesosphere playing a crucial role, on the temperatures in the summer mesopause regions is larger than the effect of processes in the summer stratosphere."*

- The authors may consider to include the first part of section 3 in a new section 3.1, and rename the old section 3.1 as section 3.2 (for a matter of symmetry in the presentation).

This has been done.

**Second Review of "The role of the winter residual circulation in the summer mesopause regions in WACCM" by Maartje Kuilman and Bodil Karlsson (acp-2017-647).**

The authors have implement the reviewer comments well and thus increased the readability and plausibility of their arguments. Therefore, I recommend publication after processing of the following minor and mostly technical comments.

First of all, we would like to thank the reviewer for their positive assessment, their constructive criticism, and time spent to analyze our manuscript again. We are grateful for the valuable suggestions provided. Responses to each of the comments are listed below.

Minor comments:
o Line 654: …relatively freely up into the mesosphere…
This has been changed.

o Line 752 – 754: Please remove this sentence, since NLC are not a part of the paper anymore
This has been removed.

o Line 791: is based on …
This has been changed.

o Line 794: In this study, the F_200_WACCM …
This has been changed.

o Line 829 – 833: Here the assumption is made that the altitude of the zonal wind reversal is correct in WACCM. However, it is not \cite. Please mention that fact.

We are interested in the difference between the cases with and without GWs in the winter hemisphere. The exact values of the wind reversal might not be captured in WACCM exactly right, but that wouldn't change our arguments. For the zonal wind in WACCM see Figure 5-7. Lines 829-833 have now been rewritten.

The figure below shows that the positive meridional heat flux in the midlatitude winter hemisphere is indeed distributed over a wider altitude range.

[Figure]

[Figure]

o Line 845 – 847: Just a curious question: What is about the meridional temperature gradient when the summer polar mesopause warms as well?

I am not quite sure what the reviewer is asking for: the meridional temperature gradient in which region? I include the figures that should answer the question.

[Figure]

Fig. 3 shows the temperature anomalies for high (left) and low (right) planetary wave activity, as measured by the temperature in the winter stratosphere (1-10 hPa, 60°S-40°S) in July for the control run (first row) and run without GWs in the winter hemisphere (second row). The dotted areas are regions where the correlation has a p-value < 0.05. The black 150 K-contour indicates the polar mesopause region.

These were the anomaly fields, for the absolute temperatures, see the following figure:

[Figure]

Temperature July (left) and temperature July when the GWs in the NH off (right).

o Line 851: … NLC region is about 210-22K, which is much higher …
This has been changed.

o Line 858: They attributed that the warming …
This has been changed.

o Line 890: …and in the summer mesopause region outside the polar region. -> The correlation is weak and not significant in the polar mesosphere (see Fig. 2 and 4).

It is true that the temperature anomalies in the summer polar mesosphere not in all the cases reaching a confidence level of 95% all the way to the poles. We comment on this in line 382-387:

*" This is in agreement with the results presented in Karlsson et al. (2009) although the WACCM temperature response does not reach statistical significance at a 95% level all the way to the polar region. This could be due to time lags between the response in the summer mesopause and the dynamic activity in the winter: Karlsson et al. (2009) found a lag between the winter and the summer hemisphere of up to 15 days. In the monthly-mean approach that we use for this study, lags in time are not accounted for."*

o Line 939: … change due to turning off the winter GWs …
This has been changed.

o Line 971: In Fig. 2, it is seen …
This has been changed.

o Line 973: … in the NH summer polar mesosphere. -> the correlation in the polar mesosphere is weak and not significant

It is true that the temperature anomalies in the summer polar mesosphere not in all the cases reaching a confidence level of 95% all the way to the poles. We now speak about the summer mesosphere.

o Line 976: …temperature in the summer polar mesosphere. -> Same as above.
See earlier comment.

o Line 981: … is a change in the stratospheric meridional temperature gradient…
This has been changed.

o Line 987: This can been shown clearly …
This has been changed.

o Line 1020 -2024: The statement of this sentence is not clear.

I assume the reviewer means line 1020-1024. This sentence has now been rewritten.

*" We conclude that for both hemispheres, the effect of PW activity on the summer polar mesosphere temperatures would be the opposite, if changes in the summer stratosphere were acting alone. Hence, the IHC as described by e.g. Karlsson et al. (2009) still holds as the dominant mechanism governing the monthly mean temperatures variability in the summer polar mesosphere, at least for July."*

o Line 1033: … has a net cooling effect on the summer polar mesosphere differing in magnitude between the two hemispheres.
This has been changed.

o Line 1046: …: in this case a weak BDC leads to cooling of the summer mesosphere region.
This has been changed.

o Line 1047 – 1049: please rewrite this sentence to make it clearer that the net effect of the IHC is probably larger than that of the intrahemispheric coupling from the stratosphere of the summer hemisphere.
The conclusion has now been rewritten to make our points more clear.

Additionally, it should be mentioned in the conclusion that the positive correlation between the winter stratosphere and summer mesosphere only barely reaches the polar mesopause region
This has now been mentioned.

[revised manuscript text omitted]